# Relating Implicit Bias and Adversarial Attacks through Intrinsic Dimension

## Abstract

Despite their impressive performance in classification, neural networks are known to be vulnerable to adversarial attacks. These attacks are small perturbations of the input data designed to fool the model. Naturally, a question arises regarding the potential connection between the architecture, settings, or properties of the model and the nature of the attack. In this work, we aim to shed light on this problem by focusing on the implicit bias of the neural network, which refers to its inherent inclination to favor specific patterns or outcomes. Specifically, we investigate one aspect of the implicit bias, which involves the essential Fourier frequencies required for accurate image classification. We conduct tests to assess the statistical relationship between these frequencies and those necessary for a successful attack. To delve into this relationship, we propose a new method that can uncover non-linear correlations between sets of coordinates, which, in our case, are the aforementioned frequencies. By exploiting the entanglement between intrinsic dimension and correlation, we provide empirical evidence that the network bias in Fourier space and the target frequencies of adversarial attacks are closely tied.

## 1 Introduction

An active field of research in artificial neural networks (ANNs) is focused on understanding why, despite their enormous success, their predictions can be drastically changed by subtle perturbations of their inputs, known as adversarial attacks (Szegedy et al., 2013). New research has shown a strong correlation between the implicit bias of artificial neural networks - which refers to their natural predisposition to exhibit a preference towards particular patterns or results - and their ability to resist adversarial attacks. This was highlighted in a recent study (Faghri et al., 2021), wherein it was demonstrated that the specific optimizer, neural network architecture, and regularizer employed had a substantial impact on the ability of a linear neural network to withstand adversarial interference. However, besides simple models (Gunasekar et al., 2018), a formal characterization of the implicit bias of a neural network remains a formidable challenge. The research presented in Karantzas et al. (2022) offers an algorithm aimed at investigating a specific aspect of implicit bias even in the case of complex networks. This approach involves analyzing the *essential* input frequencies required to maintain the accuracy of a trained network. Such frequencies are computed by training, for each input image, a learnable modulatory mask that filters the frequency content of the image, reducing it to the bare minimum required to preserve correct classification. The essential frequency masks can serve as a unique fingerprint for the network, as they encapsulate the information that the ANN relies on when processing inputs.

In this work, we leverage this methodology to investigate the correlation between the implicit spectral bias of the network, defined in terms of the image frequencies that are essential to perform the correct classification, and the frequencies targeted by adversarial attacks to deceive the network. In particular, for each image, we calculate the modulatory mask of the essential frequencies (using a similar approach to Karantzas et al. (2022)) and, additionally, for the same image, we learn a mask containing the essential adversarial frequencies needed for an attack to be successful. Fig. 1 displays examples of clean and attacked images before (A, B) and after (C, D) being filtered by, respectively, essential frequency masks and adversarial frequency masks.

We use these two sets of masks to check the dependence (or lack thereof) between the network bias in the Fourier domain and the frequencies that are being targeted by the adversarial attack. Our

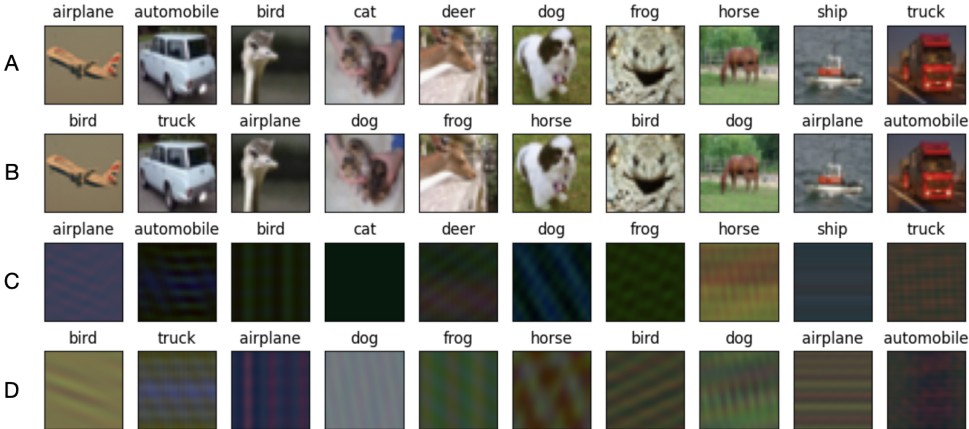

Figure 1: Examples of CIFAR-10 (Krizhevsky, 2009) images before and after being filtered by the Fourier masks: (A): original input images (B): adversarial images generated with $\ell_\infty$ Fast Minimum Norm (Pintor et al., 2021) attack on ResNet-20 (He et al., 2016) (C): images filtered by essential frequency masks (D): adversarial images filtered by adversarial frequency masks.

primary objective is to offer empirical proof that the network spectral bias determines the nature of the adversarial attacks in Fourier space, in the same spirit of Faghri et al. (2021). However, defining and computing this correlation is a challenging task due to the high-dimensional nature of the modulatory mask sets, and the fact that their correlation can be, in principle, highly non-linear. To address these challenges we introduce a novel non-linear correlation method that relies on the observation that the intrinsic dimensionality ($I_d$) of a data set is affected by correlations between the features. By comparing the $I_d$ estimated in the data set with the distribution of $I_d$ that one would obtain in the case of fully uncorrelated data, we are able to quantify the probability that the two types of masks are correlated. Our findings indicate a strong correlation between the feature spaces defined by the two types of masks, providing empirical evidence of the connection between network bias in Fourier space and target frequencies of adversarial attacks.

## 2 RELATED WORK AND BACKGROUND

### 2.1 IMPLICIT BIAS AND IMPLICIT FOURIER BIAS

The idea behind the phenomenon of implicit bias is that the loss landscape of an overparameterized network has many local minima, and which local minimum one converges to after training depends on the complex interplay between factors including the choice of the model architecture and parameterization (Gunasekar et al., 2018; Yun et al., 2020), the initialization scheme (Sahs et al., 2022), the optimization algorithm (Williams et al., 2019; Woodworth et al., 2020) and the data statistics (Yin et al., 2019). The implicit bias of state-of-the-art models has been shown to play a critical role in the generalization property of deep neural networks (Li et al., 2019; Arora et al., 2019). Analytical characterizations of the implicit bias have been provided only for deep linear convolutional or fully connected networks (Gunasekar et al., 2018). One interesting effect of the implicit bias of the network is its tendency to learn specific frequencies in the *target function* during training, a phenomenon called spectral bias (Rahaman et al., 2019). This bias results in the network learning low complexity functions and can potentially explain its ability to generalize (Fridovich-Keil et al., 2022; Cao et al., 2019; Wang et al., 2020; Tsuzuku & Sato, 2019). Also, not surprisingly, the implicit bias strongly influences the type of input features extracted by a trained neural network. In particular, in Karantzas et al. (2022), the authors show that very few image frequencies in the Fourier domain are essential to the network to perform classification. These findings have helped to characterize the spectral bias of neural networks with a focus on the input space rather than the target function (as in Rahaman et al. (2019)).

Interestingly, a deep connection exists between robust classification and implicit bias (Faghri et al., 2021). Empirically, a strong relationship has been found between the network robustness and the

statistics of the Fourier spectra of the input data (Yin et al., 2019) or architecture (Caro et al., 2020) and detection strategies in the Fourier domain have been used to defend against adversarial attacks (Harder et al., 2021).

## 2.2 ADVERSARIAL ATTACKS

Artificial Neural Networks are well known to be vulnerable to adversarial attacks (Szegedy et al., 2013). These attacks involve manipulating an input data point in a way that deceives an otherwise well-performing classifier, by making small alterations to a correctly classified data point. Numerous techniques have been proposed to create such adversarial examples, beginning with the Fast Gradient Sign Method (FGSM) (Goodfellow et al., 2014), followed shortly by variants such as Projected Gradient Descent (PGD) (Madry et al., 2018). Both these methods employ gradient information to generate an appropriate adversarial example while ensuring that the $\ell_p$ norm of the perturbation remains below a fixed threshold $\epsilon$. These algorithms were primarily developed for effectiveness rather than optimality, which may limit their ability to generate input samples with minimal perturbations, resulting in them being classified as "maximum confidence" attacks. In contrast, "minimum norm" attacks prioritize the identification of adversarial examples with the least amount of perturbation by minimizing its norm. In this regard, some of the most notable proposals are L-BFGS (Szegedy et al., 2013), the Carlini and Wagner attack (Carlini & Wagner, 2017), DeepFool (Moosavi-Dezfooli et al., 2015) and the recent Fast Minimum Norm (FMN) attack (Pintor et al., 2021), which seeks to combine the efficiency of FGSM and PGD with optimality in terms of perturbation norm.

The robustness of neural networks against adversarial attacks remains an unresolved issue. Although adversarial training is currently the most effective technique for improving the resilience of neural classifiers, it often involves a trade-off between robustness and a reduction in performance on non-adversarial, clean data (Goodfellow et al., 2014). Moreover, it remains unclear why adversarial examples exist and whether they represent an inevitable byproduct of current neural architectures and training methods (Ilyas et al., 2019; Shafahi et al., 2019). The goal of this work is not to propose a method for improving the adversarial robustness of neural networks. Rather, our aim is to provide valuable insights into the frequency content that is targeted by adversarial attacks and its relationship with the implicit spectral bias of the network.

## 2.3 INTRINSIC DIMENSION

The concept of the intrinsic dimension ($I_d$) of a data set is widely used in data analysis and Machine Learning. Before providing a more formal definition, imagine a data set where your data points are the cities around the globe described by their 3D Cartesian coordinates. We will say that the *embedding dimension* of this data set is three. However, anyone familiar with cartography would agree that nearly the same information can be encoded with only two coordinates (latitude and longitude). Therefore, its $I_d$ would be equal to two. Indeed, one of the definitions of $I_d$ is the minimum number of coordinates needed to represent the data with minimal information loss. A complementary definition is the dimension of the manifold in which the data lies, that in this case would be a sphere.

The intrinsic dimension estimation is closely related to the field of dimensionality reduction since it gives a hint about which should be the dimension of the projection space to avoid information loss. Thus, one possible way of estimating the $I_d$ is to find a meaningful projection into the lowest dimensional space possible. A classical method for doing that is Principal Component Analysis (Wold et al., 1987), but it has the drawback that, strictly speaking, it is only correct if the data lie in a hyperplane, since it performs a linear transformation. Therefore, the development of methods for overcoming such a limitation is an active research field, resulting in techniques like Multidimensional Scaling (Borg & Groenen, 2005), Isomap (Balasubramanian & Schwartz, 2002), t-distributed stochastic neighbor embedding (t-SNE) (van der Maaten & Hinton, 2008) or Uniform Manifold Approximation and Projection (UMAP) (McInnes et al., 2018), to mention some. Other methods can estimate the $I_d$ of a data set even in the case in which projecting in the lower dimensional space is not possible (for example, due to topological constraints). Typically, these approaches infer the $I_d$ from the properties of the Nearest Neighbors' distances. While a full review of these methods is out of the scope of this work (the interested reader is referred to Lee et al. (2015)), it is worth mentioning the Maximum Likelihood approach (Levina & Bickel, 2005), the Dimensionality from Angle and

Norm Concentration (DANCo) approach (Ceruti et al., 2014) or the two-NN (Facco et al., 2017). The last is the one employed in this work since it is particularly fast and it behaves well even in the case of data sets with a high non-uniformity on the density of points.

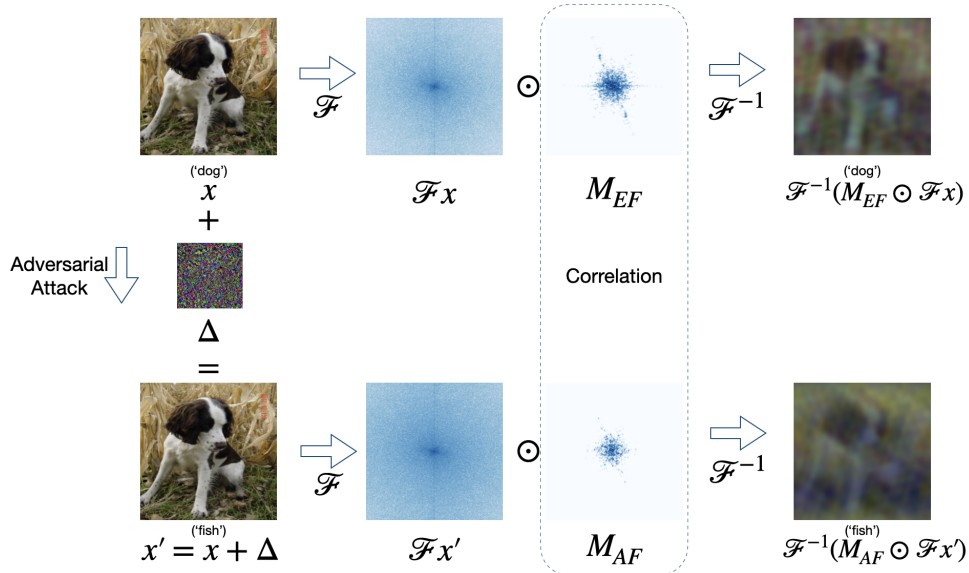

Figure 2: Schematic representation of the method employed to obtain essential frequency masks and adversarial frequency masks. Only one channel is displayed for visualization purposes. Full details are provided in Sec. 4.4.

## 3 METHODS

### 3.1 MODULATORY MASKS

The primary tools we use to gather insights on the implicit spectral bias and on the geometry of adversarial examples are modulatory masks. The latter retain information on the essential frequencies required to achieve a particular classification task. To obtain these masks, we follow a similar algorithm to the one outlined in Karantzas et al. (2022), as depicted in Fig. 2. We train masks that modulate the frequency content of an image by multiplying element-wise each entry of the Fast Fourier Transform (FFT) of the image with the corresponding entry of the mask, which is a learnable scalar between 0 and 1. Specifically, starting from an image $x$, we compute its FFT $\mathcal{F}x$ and multiply it element-wise with a learnable mask $M$. The mask has the same shape of the image $x$ (and its FFT $\mathcal{F}x$), meaning that if the image has RGB encoding we train a separate mask for each channel, and its entries are constrained to be in $[0, 1]$. The result of this multiplication is then projected back in pixel space by taking the real part of its inverse Fourier transform, thereby obtaining a new filtered image $x_F$:

$$x_F = \Re(\mathcal{F}^{-1}(M \odot \mathcal{F}x)). \tag{1}$$

The image $x_F$ is then fed into the trained classification model to obtain a prediction. We produce two sets of masks. The masks belonging to the first set encode the essential frequencies of an image to be correctly classified by the neural classifier, thus we will refer to these as *essential frequency masks* ($M_{EF}$). The second set is composed of masks that encode the essential frequency content required to maintain the effectiveness of an adversarial attack, that is, the essential frequencies needed to misclassify an adversarially perturbed image. We will refer to these masks as *adversarial frequency masks* ($M_{AF}$). Some examples of adversarial frequency masks are shown in Fig. 3 (the corresponding $M_{EF}$ masks are shown in the Appendix in Fig. 6). Both sets of masks are learned using a preprocessing layer attached to a classifier ANN with frozen parameters. The essential frequency masks are trained by optimizing the Cross-Entropy loss of the entire model (consisting of the

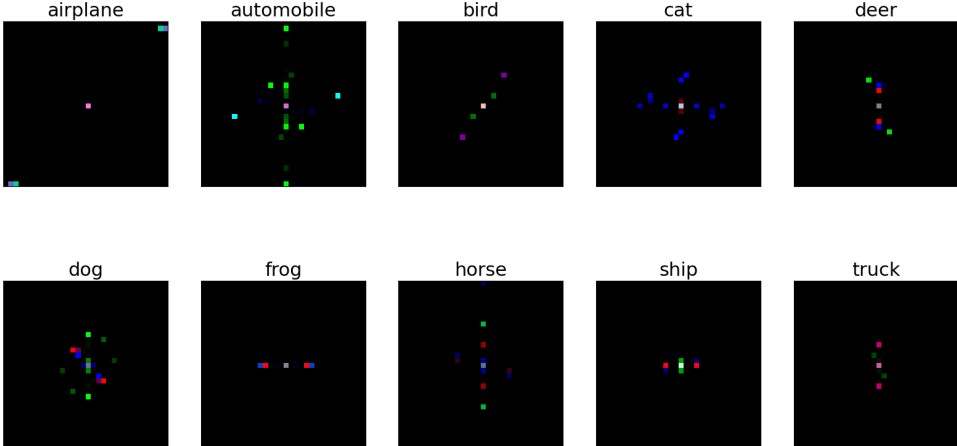

Figure 3: Examples of adversarial frequency masks, represented as RGB images. The labels refer to the classification of the clean image. The masks were obtained using CIFAR-10 and the Fast Minimum Norm attack on ResNet-20.

preprocessing layer and the trained classifier) on the original samples. Conversely, for adversarial frequency masks, the training objective is the Cross-Entropy with respect to the adversarial class (to preserve misclassification), and the masks are trained on adversarial data. The key property of the learned masks is their *sparsity*, which is achieved by enforcing an $\ell_1$ norm regularization on the entries of the mask during training. This regularization ensures that the mask accurately captures only the essential frequency content needed to accomplish a specific task, such as correctly classifying an input or misclassifying an adversarial example. Our primary objective is to determine whether a correlation exists between these distinct sets of masks. To do so, we propose a novel algorithm based on intrinsic dimension estimation. This algorithm overcomes the limitations of existing methods and is applicable to non-linearly correlated data.

### 3.2 Non-linear correlation through Intrinsic Dimension

As mentioned earlier, to examine the statistical relationship between implicit bias and adversarial attacks, it is necessary to compute correlations between two feature spaces that characterize the same images: the essential frequencies for image classification and those required for the adversarial attack to be successful. The conventional approach for investigating correlations is based on the Pearson correlation coefficient ($R^2$) between variables (Pearson, 1896). However, this method has two limitations that make it impractical. First, it cannot be applied to assess correlations between two sets of multiple variables, such as the different types of masks mentioned earlier. Second, it is unable to detect non-linear correlations, as illustrated in the example presented in Fig. 4. Therefore, we provide a new approach that overcomes these problems by using the intrinsic dimension.

The intrinsic dimension of a data set is closely linked to the correlations among the various features that define the data points. These correlations determine the regions in which the data points can exist, thereby shaping the underlying manifold. As previously mentioned, the dimension of this manifold corresponds to what we refer to as the intrinsic dimension. Let us consider the simplest example: a two-dimensional data set. If the two variables are uncorrelated, the correlation coefficient ($R^2$) approaches zero while, if one feature is a linear function of the other, $R^2$ becomes equal to one. In the context of the data manifold, the first scenario corresponds to a plane ($I_d = 2$), while the second scenario corresponds to a line ($I_d = 1$). However, if we consider a slightly more complex scenario, the advantage of using the $I_d$ becomes evident. The spiral data set in Fig. 4 has $R^2 \approx 0$ due to the non-linear nature of the correlation between the two variables, while the behavior of the $I_d$ is identical to the one observed on the linearly correlated data set. Moreover, there is no theoretical limit to the dimension of the data sets for which it can be computed. Hence, we employ an approach in which we assess the probability that the observed intrinsic dimension ($I_d$) is consistent with the intrinsic dimension that would be measured if both sets of coordinates were entirely uncorrelated. It involves four steps (illustrated in Fig. 4):

1. Estimating the intrinsic dimension ($I_d$) of the combined data set, obtained by concatenating the two sets of variables.

2. Generating multiple fully uncorrelated data sets by shuffling the positions of data points within one of the two sets of coordinates.

3. Estimating the average and standard deviation of the intrinsic dimension ($I_d$) for the uncorrelated data sets.

4. Applying a one-sided $Z$-test to determine the probability that the intrinsic dimension ($I_d$) estimated in step 1 is significantly lower than the average estimated in step 3.

The key step enabling the usage of the $I_d$ to detect correlations is the second one, where we shuffle one of the two coordinate sets so that every vector belonging to the first set gets paired with a randomly chosen vector of the second set. By shuffling the order of the data points, the probabilities of the two sets of coordinates $p\left(\mathbf{x_1}\right)$ and $p\left(\mathbf{x_2}\right)$ remain unaltered but the joint probability becomes, by construction, $p\left(\mathbf{x_1}, \mathbf{x_2}\right) = p\left(\mathbf{x_1}\right) p\left(\mathbf{x_2}\right)$. However, this will not be the case if there is a correlation between $\mathbf{x_1}$ and $\mathbf{x_2}$ (see Fig. 5 in the Appendix for an example). Therefore, by examining the joint probability distribution before and after shuffling, we can discern whether there exists a correlation between $\mathbf{x_1}$ and $\mathbf{x_2}$. As explained above, this method overcomes the difficulties inherent in finding non-linear correlations between sets of coordinates. The $Z$-test may be limited as it assumes normality in the distribution of computed $I_d$ values on the dataset with shuffled coordinates. While this is generally fulfilled in the cases studied here, a more significant challenge arises due to the curse of dimensionality. The number of points needed to estimate the $I_d$ with a given level of accuracy increases nearly exponentially with the $I_d$ (Bac et al., 2021), making it challenging for datasets with high intrinsic dimension and a moderate number of points

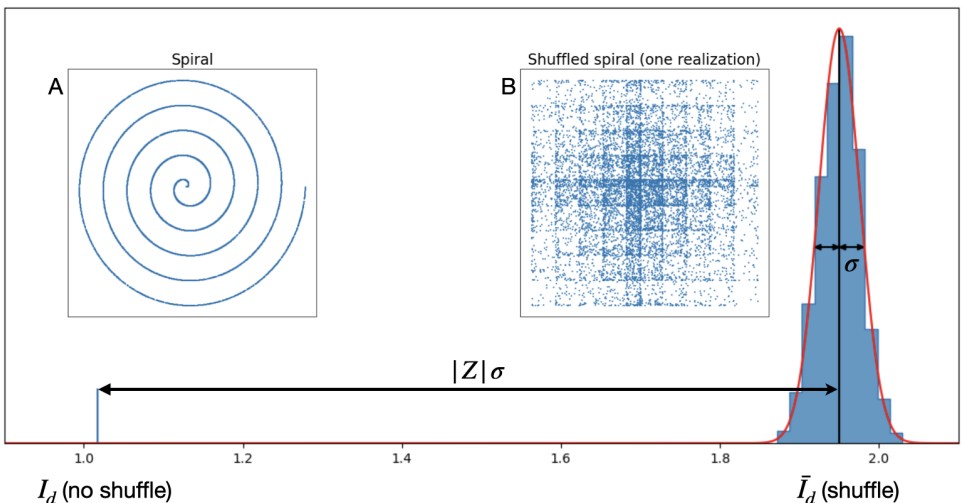

Figure 4: Schematic depiction of our proposed $I_d$-based correlation method on synthetic, spiral-shaped data. We compare the $I_d$ of the original data set (A) with the $I_d$s of the shuffled data set (B) and $Z$-test the hypothesis that the original $I_d$ is lower than the shuffled $I_d$s.

Table 1: Correlation in spiral-shaped data. ($\mathbf{R^2}$): linear correlation coefficient; ($\mathbf{I_d}$): intrinsic dimension of the spiral; ($\mathbf{I_d}$ (shuffle)): mean $\pm$ standard deviation of the intrinsic dimension of the data set obtained by shuffling one of the two coordinates; ($\mathbf{Z}$): $Z$-score for the hypothesis that the original $I_d$ is significantly lower than the average of the shuffled distribution; ($\mathbf{P\text{-value}}$): significance of the $Z$-test.

| $\mathbf{R^2}$ | $\mathbf{I_d}$ | $\mathbf{I_d}$ (shuffle) | $\mathbf{Z}$ | $\mathbf{P\text{-value}}$ |
|---|---|---|---|---|
| $2.5 \cdot 10^{-3}$ | 1.02 | $1.95 \pm 0.03$ | $-74.51$ | 0 |

## 4 EXPERIMENTAL RESULTS

### 4.1 DATA

For our experiments, we primarily utilized CIFAR-10 (Krizhevsky, 2009), a widely-used benchmark data set that consists of 60000 RGB $32 \times 32$ training images and 10000 test images categorized into 10 classes. When studying adversarial examples, we used the test images, and the training set was solely employed for fine-tuning models, as explained in greater detail in the subsequent section. We also explored the feasibility of scaling up our experiments to a higher-dimensional data set. In particular, we trained masks on Imagenette (Howard, 2022), a 10-class subset of ImageNet (Deng et al., 2009), and report the results on such data set along with the specific setup details in Sec. 4.6. However, for the majority of our analyses we relied on CIFAR-10 as the time needed to compute the intrinsic dimension of higher-dimensional mask data sets made it impractical to conduct multiple repeated runs.

### 4.2 MODELS

To gain a more accurate understanding of how our proposed method behaves in various scenarios, we employed two classification models based on different neural architectures. The first one is ResNet-20, a relatively small representative of the very well known ResNet family, introduced in He et al. (2016). The second model belongs to the class of Vision Transformers (ViT) (Dosovitskiy et al., 2021). Namely, we used CCT-7, a Compact Convolutional Transformer (Hassani et al., 2021) model, that differs from the original ViT because it employs convolutions in the tokenization phase and a smaller hidden size, which allows scaling a ViT-like architecture to small size data sets such as CIFAR-10. Training details for all the models we employed are reported in the Appendix (Sec. A.3).

### 4.3 ATTACKS

We employed the $\ell_\infty$ version of the Fast Minimum Norm (FMN) attack algorithm as our reference adversarial attack method (Pintor et al., 2021). This choice was primarily driven by the simplicity and effectiveness of the algorithm, as it does not require parameter fine-tuning and is capable of generating high-quality adversarial examples swiftly. Additionally, we conducted tests using other adversarial attack techniques, namely Projected Gradient Descent (PGD) (Madry et al., 2018) and DeepFool (Moosavi-Dezfooli et al., 2015), both in their $\ell_\infty$ versions. All the attacks were employed in the untargeted setting. For PGD, we selected a perturbation magnitude of $\epsilon = 0.01$, which was chosen to maintain consistency with the perturbation magnitude produced by the FMN attack. We provide an analysis of the robustness of our findings with respect to $\epsilon$ in the Appendix (Sec. A.8). To implement these attack algorithms, we utilized the Foolbox library (Rauber et al., 2020; 2017).

### 4.4 MASK TRAINING

The key step in our experimental procedure is the training of Fourier masks (see Sec. 3.1). Starting from a trained, well-performing classifier, we freeze its parameters and prepend to it a pre-processing layer that computes the FFT of an image, multiplies it element-wise by the trainable mask and computes the inverse FFT. The real part of the resulting image is then fed into the classifier. The process of training the masks is identical for both the set of essential frequency masks and adversarial frequency masks, with the only difference being the data set used for mask training. We train essential frequency masks using clean images associated with their original labels. In contrast, for adversarial frequency masks, we utilize adversarial images and the adversarial labels produced by the classifier for those images. In this step, we optimize the standard Cross-Entropy loss function with the addition of an $\ell_1$ penalty term to promote mask sparsity. Further details on the mask training procedure are reported in Sec. A.4 in the Appendix.

### 4.5 CORRELATION BETWEEN MASKS

To provide evidence of the relation between the implicit bias of the network and the adversarial perturbations, we adopt a direct approach: we correlate the essential frequency masks with the adversarial frequency masks. This correlation analysis is performed using our novel $I_d$-based corre-

Table 2: Correlation between essential frequency masks and adversarial frequency masks (CIFAR-10).

| Attack | Model | Cosine sim | $I_d$ | $I_d$ (shuffle) | Z | P-value |
|--------|-------|-----------|-------|----------------|-----|---------|
| FMN | ResNet-20 | $0.25 \pm 0.16$ | 31.65 | $34.98 \pm 0.73$ | $-4.56$ | $2.5 \cdot 10^{-6}$ |
|  | CCT-7 | $0.22 \pm 0.17$ | 22.93 | $24.47 \pm 0.34$ | $-4.50$ | $3.4 \cdot 10^{-6}$ |
| PGD | ResNet-20 | $0.22 \pm 0.15$ | 32.35 | $36.18 \pm 0.72$ | $-5.31$ | $5.4 \cdot 10^{-8}$ |
|  | CCT-7 | $0.21 \pm 0.17$ | 23.30 | $24.52 \pm 0.39$ | $-3.13$ | $8.7 \cdot 10^{-4}$ |
| DeepFool | ResNet-20 | $0.25 \pm 0.15$ | 30.35 | $33.93 \pm 0.73$ | $-4.91$ | $4.5 \cdot 10^{-7}$ |
|  | CCT-7 | $0.20 \pm 0.16$ | 23.44 | $25.10 \pm 0.35$ | $-4.81$ | $7.4 \cdot 10^{-7}$ |

lation method. The outcomes of our evaluation, including the results of the $Z$-test (see Sec. 3.2) and the mean cosine similarity between the masks (which serves as a linear benchmark), are presented in Table 2. To determine the $I_d$ values, we utilized the implementation of TwoNN (Facco et al., 2017) contained in the DADApy (Glielmo et al., 2022) library, on the data set generated by concatenating the essential frequency masks and the adversarial frequency masks. We then compare these $I_d$ values with the distribution of $I_d$ obtained by shuffling the order of one of the two sets of masks (performing the shuffling process 50 times for each setup). We employ a one-sided $Z$-test to assess the hypothesis that the original $I_d$ value is significantly lower than the average of the shuffled $I_d$s. For all models and attacks tested, our findings indicate a significant correlation between the two sets of masks.

## 4.6 Correlation results on Imagenette

To further evaluate our approach, we conducted experiments on a 10-class subset of the ImageNet data set (Howard, 2022). The subset consisted of 9469 training samples and 3925 test samples, which were resized to $224 \times 224$. We employed a ResNet-18 (He et al., 2016) classifier and conducted the training of modulatory masks (essential frequency masks and adversarial frequency masks) according to the same procedure outlined in Sec. 4.4 for CIFAR-10, with the only difference that both the training images and test images were used to calculate the masks. We made this choice because the accurate estimation of intrinsic dimension is crucial for our $I_d$-based correlation method (see Sec. 3.2), and the number of data points needed for reliable estimation scales exponentially with the intrinsic dimension (Bac et al., 2021). Being significantly higher-dimensional than CIFAR-10, the Imagenette data set yields noticeably higher $I_d$ values on the modulatory masks. Hence, relying solely on the smaller test set would have been insufficient, leading us to the decision to augment it with the training images.

We conducted correlation tests between essential frequency masks and adversarial frequency masks using our $I_d$-based method, and the results are summarized in Table 3. The probability of correlation is high for FMN and DeepFool attacks, with $P$-values of the $Z$-test in the order of $10^{-2}$. However, it is important to note that the estimation of $I_d$ may have been compromised by the scarcity of data points, as indicated by the high variance in the measurements. In the case of PGD attack, the intrinsic dimension reached values well above 80 in the non-shuffled data set, which further hampered the accuracy of $I_d$ estimation. Consequently, the results obtained with this number of points are not considered reliable. To address this issue, the most straightforward approach is to increase the size of the data set used for mask generation. In this regard, we evaluated the possibility of further upscaling our experiments to the full ImageNet ILSVRC 2012 data set, as it contains 50000 images in the validation set alone. However, despite having computed modulatory masks for such data, we found out that repeated $I_d$ computation on such an amount of data becomes infeasible both in terms of memory and time requirements.

## 4.7 Class-specific content in masks

Expanding upon the findings presented in Karantzas et al. (2022) regarding the clustering of modulatory masks, we propose a hypothesis that masks computed on images of the same class possess similar frequency content. To validate this hypothesis, we designed a simple test, whose results are displayed in the Appendix in Sec. A.6. We applied multiple times our $I_d$-based correlation method

Table 3: Correlation between essential frequency masks and adversarial frequency masks (Imagenette, ResNet-18).

| Attack | Cosine sim | $I_d$ | $I_d$ (shuffle) | Z | P-value |
|---|---|---|---|---|---|
| FMN | $0.22 \pm 0.10$ | 65.06 | $69.18 \pm 2.12$ | $-1.94$ | $2.6 \cdot 10^{-2}$ |
| PGD | $0.12 \pm 0.07$ | 81.80 | $77.25 \pm 2.94$ | $1.54$ | $9.4 \cdot 10^{-1}$ |
| DeepFool | $0.15 \pm 0.09$ | 65.14 | $69.90 \pm 2.50$ | $-1.90$ | $2.8 \cdot 10^{-2}$ |

to subsets containing $k$ randomly chosen classes, with $k$ ranging from 1 to 10. If masks belonging to the same class shared common frequencies, we would anticipate the average P-values to decrease (and, consequently, correlation probability to increase) as we added more classes. This is because increasing the number of classes would decrease the probability of matching masks belonging to the same class when they are shuffled. In the experimental results illustrated in Fig. 7, a distinct downward trend in P-values can be observed as $k$ increases, indicating that there is a considerable amount of class-specific information present in the masks.

Based on this observation, we envisioned the possibility of training a single mask that encodes the essential frequency content for an entire class. Such masks (one for each class) can be obtained following the same approach used to learn essential frequency masks for single images, but training on all the images belonging to a certain class. We trained class-level masks on the training images of CIFAR-10 on ResNet-20 and observed that they effectively preserved correct classifications for the unseen test set. Even more interestingly, we noted that these class-level essential frequency masks also successfully mitigated the impact of adversarial attacks on most of the images. Quantitative results for this analysis are detailed in the Appendix (Sec. A.7). While this discovery alone is insufficient for constructing an adversarial defense technique, as countering the attack necessitates knowledge of the correct class to select the corresponding mask, we believe it represents a promising starting point for future research in this direction.

## 5    DISCUSSION

Our study delves into the relationship between adversarial attacks and the implicit bias of neural networks. We introduce a novel method to uncover non-linear correlations, revealing a link between the minimum frequency content needed for correct image classification and adversarial attack frequencies. The analysis covers standard network architectures like ResNets and ViTs and data sets such as CIFAR-10 and Imagenette. This work represents a significant advancement in understanding the relationship between the implicit bias of neural networks and their robustness properties, in the same spirit of Faghri et al. (2021) but for models where the implicit bias is not available in an explicit form. Our results hold prospective implications for the field of adversarial attacks: the deceptive nature of these data manipulations is not yet fully comprehended, and our findings shed light on the crucial frequencies utilized by attackers. This understanding has the potential to drive the development of new defense and detection algorithms, enhancing the security and robustness of neural networks. Furthermore, our mask-based approach offers the ability to modulate both the phase and modulus in the Fourier transform of the data opening up new avenues for investigating the implicit frequency bias of a network. By manipulating these data features, we can gain deeper insights into the implicit bias and explore the influence of different frequency components on classification outcomes. In addition, other types of representations, such as wavelets, could be explored.

Finally we note that the method employed in this paper for discovering non-linear correlations between feature spaces, based on $I_d$, exhibits intriguing potential applications beyond the scope of this study. Correlations play a vital role in various scientific domains, including physics (Gallus et al., 2023), economics (Fleckinger, 2012), epidemiology (Majumder & Ray, 2021), and social networks (Starnini et al., 2017), among others. Therefore, it would be interesting to examine whether this method can unveil correlations that were previously unseen using conventional approaches. To these aims, a theoretical development that explores the relationship between $I_d$ and conventional methods for addressing this problem is valuable. Such an investigation could lead to possible enhancements that either overcome the limitations of the method or enable more precise quantification of correlation strength. These research directions form part of our future objectives.

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

# A APPENDIX

## A.1 COMPUTATIONAL RESOURCES

In terms of computational requirements, our experimental procedure encountered two major resource-intensive tasks. First, training masks necessitated individual training for each image, which consumed substantial computational resources. To expedite this process, we utilized hardware acceleration and performed the training on a NVIDIA V100 GPU. Second, the repeated computation of intrinsic dimension estimates on large, high-dimensional mask data sets posed computational challenges. For this step, we employed a CPU for analysis. Specifically, we executed the analysis on either a remote AMD EPYC 7542 system or a local machine equipped with an 8-core Apple M1 chip.

## A.2 MORE DETAILS ON SPOTTING THE CORRELATION THROUGH INTRINSIC DIMENSION

As mentioned in the main text, the crucial step in testing the statistical dependence between the two types of masks using intrinsic dimension is the shuffling process. By randomly altering the order in which the two data sets are combined, this step provides a baseline for what one would expect in the case of statistical independence. Moreover, through repetition, it facilitates the use of the $Z$-test.

To illustrate the behavior of the data sets during this step, we utilize the spiral data set presented in Fig. 4 and plot the joint distributions $p(x, y)$ as well as the marginal distributions $p(x)$ and $p(y)$ (see Fig. 5). It is apparent that while the marginal probabilities remain the same for both the original and shuffled data, the joint probability differs significantly. In fact, the joint probability for the shuffled data set is the product of the marginal probabilities for each variable, as expected for statistically independent data. This differs substantially from the original data, where the two variables are non-linearly correlated.

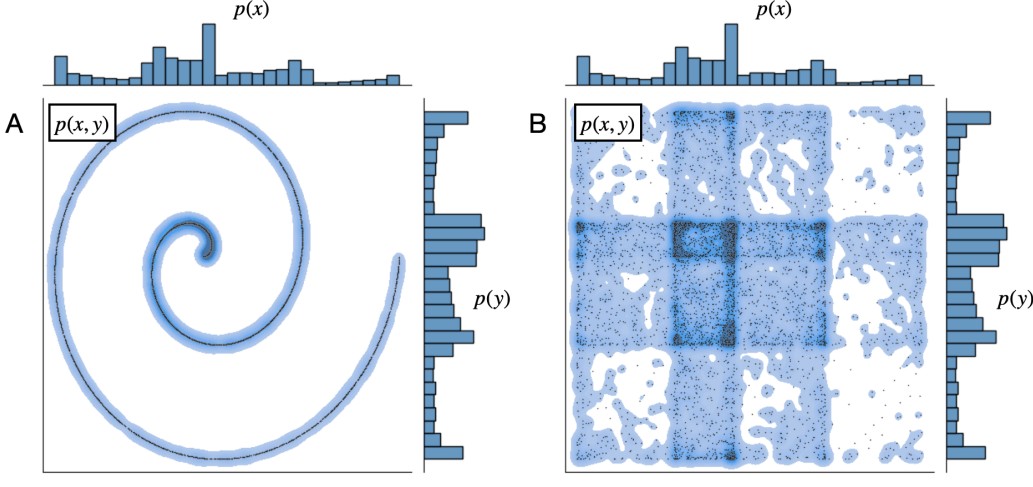

Figure 5: Joint plots for the spiral data set. Marginal distributions $p(x)$ and $p(y)$ are represented as histograms at the top and at the right of the plots. Joint distributions are estimated with kernel density estimation and represented as color maps on the $xy$ surface. In Panel (A) the columns $x$ and $y$ were concatenated using the original order of the data set, while in Panel (B) the order of column $y$ was randomly shuffled before concatenating.

## A.3 TRAINING DETAILS AND HYPERPARAMETERS

As introduced in Sec. 4, we trained two models on CIFAR-10 (Krizhevsky, 2009) (ResNet-20 (He et al., 2016) and CCT-7 (Hassani et al., 2021)) and one on Imagenette (Howard, 2022), namely a ResNet-18 (He et al., 2016). Here we provide additional details regarding the training (or fine-tuning) procedure of each model and the hyperparameters we chose.

Our ResNet-20 implementation was taken from Idelbayev (2021) and it follows the original architecture the authors of He et al. (2016) proposed for CIFAR-10. This model was trained from randomly initialized weights for 200 epochs, using Stochastic Gradient Descent with momentum, set to 0.9. The learning rate was initially set to 0.1, to be decayed by a factor 10 twice, after 100 and 150 epochs. $\ell_2$ regularization was employed, by means of a weight decay factor of $10^{-4}$. The final accuracy on the test set of CIFAR-10 was 92.23%.

For CCT-7, we fine-tuned the model for 50 epochs starting from pre-trained weights provided by the authors of the model. We used Adam (Kingma & Ba, 2015) with a fixed learning rate of $10^{-6}$, achieving a final test accuracy of 95.64% on the CIFAR-10 data set.

Finally, for ResNet-18 we started from pre-trained ImageNet weights. Since Imagenette has 10 classes, compared to the 1000 of ImageNet, we had to replace the classification head, and we trained that layer only while keeping the other layers frozen. We trained the model for 20 epochs using Adam (Kingma & Ba, 2015), with a cyclic learning rate schedule, with maximum at 0.01. We achieved a final test accuracy of 98.37%.

### A.4 DETAILS ON THE MASK TRAINING PROCEDURE

For all the modulatory masks we produced, we employed Adam optimizer (Kingma & Ba, 2015) with a learning rate of 0.01. According to a criterion similar to early-stopping, masks were trained until convergence, and in any case for no less than 500 optimization steps each. We computed modulatory masks only for images belonging to the test set of CIFAR-10 that were not previously utilized in the initial fine-tuning of the classifiers. More specifically, essential frequency masks were trained only for correctly classified images, while adversarial frequency masks were trained exclusively for correctly classified images that were successfully made adversarial by the attack. Sparsity of masks was enforced by means of $\ell_1$ regularization, weighted by a factor $\lambda = 0.01$.

The same setup was employed for training the class-level masks introduced in Sec. 4.7, with the only difference that the training procedure was conducted on the set of all images belonging to each class instead of individual images.

### A.5 EXAMPLES OF ESSENTIAL FREQUENCY MASKS

Here we provide some examples of essential frequency masks, computed on the same images employed to obtain the examples of adversarial frequency masks displayed in Fig. 3. Masks were trained starting from CIFAR-10 images on ResNet-20 architecture.

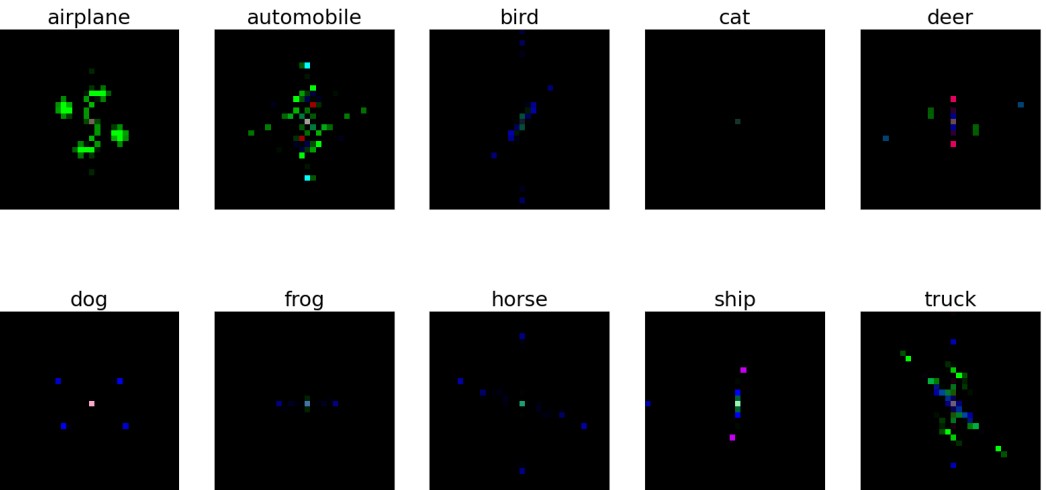

Figure 6: Examples of essential frequency masks.

### A.6 ADDITIONAL DETAILS ON DETECTING CLASS-SPECIFIC CONTENT IN MASKS

In this section we present a detailed explanation of the experiment we designed to assess the class-specificity of masks (see Sec. 4.7). Specifically, for each $k$ between 1 and 10, we sample $n = 20$ random subsets of $k$ classes, and apply our $I_d$-based correlation method to essential frequency and adversarial frequency masks belonging to that specific subset of classes. Then, for each $k$, we consider the average P-value over the $n$ random subsets we sampled. As an example, for $k = 1$ at each iteration we consider one random class alone, while for $k = 10$ the subset of classes we are considering is not random at all, and we are correlating the masks for all images in the CIFAR-10 dataset.

In case there is a strong similarity between same-class masks, we would expect our method to return higher P-values (lower signal of correlation) for small values of $k$, as the fewer classes are considered, the more likely it is that, after shuffling, an essential frequency mask gets paired with an adversarial frequency mask of the same class.

In fact, in Fig. 7 a clear decreasing trend is visible for the correlation P-value versus $k$. This trend is present with all the attacks we tested (FMN, PGD and DeepFool).

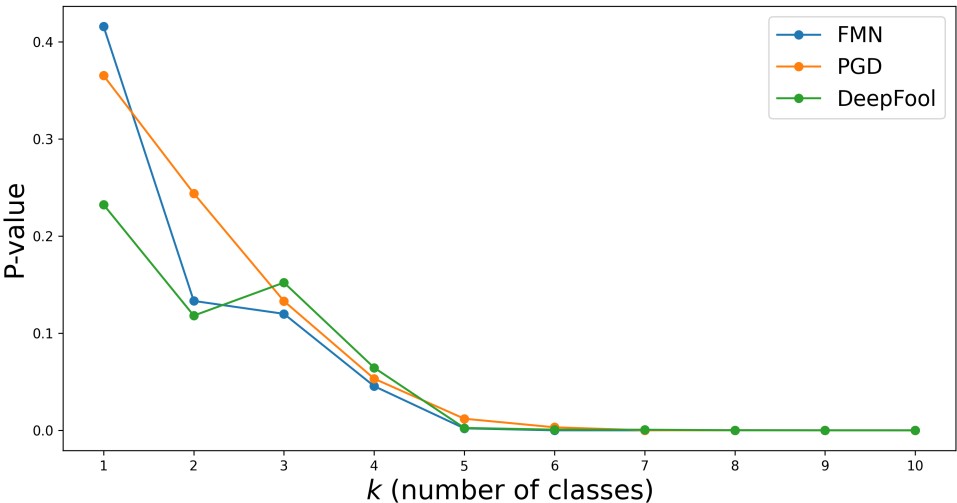

Figure 7: Average correlation P-value between $k$-class sets of essential frequency and adversarial frequency masks (ResNet-20).

## A.7 CLASS-LEVEL ESSENTIAL FREQUENCY MASKS

Here we display the class-level masks we obtained on the training set of CIFAR-10, using the ResNet-20 model. As mentioned in the main text (Sec. 4.7), these masks successfully capture the

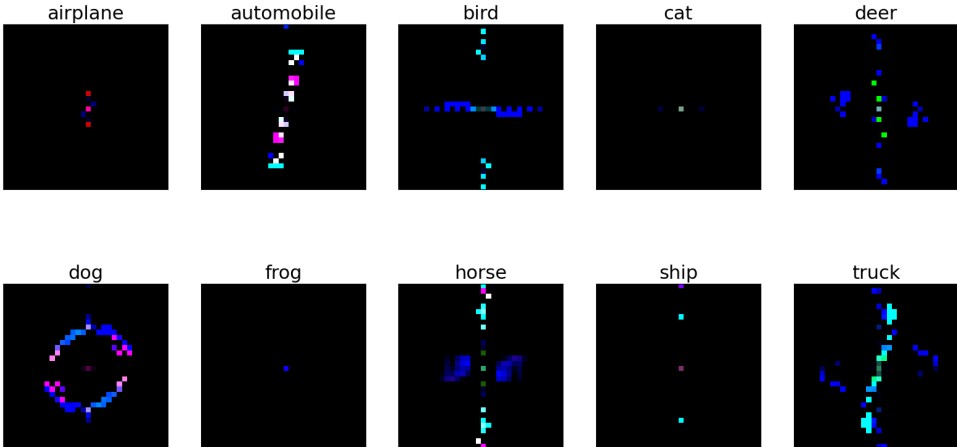

Figure 8: Class-level essential frequency masks (obtained on CIFAR-10 with ResNet-20).

essential frequency content needed to correctly classify test images, unseen during the training of the masks. Moreover, when applied to adversarial data, class-level masks are able to revert the effect of the adversarial attack in almost all cases, restoring the correct label. Table 4 reports the attack success rate (defined as the ratio of initially correctly classified images that become misclassified after the attack) for FMN, PGD and DeepFool on ResNet-20, with and without the class-level mask filter. All attacks are almost always successful, but only less than $8\%$ of the attacked images stay adversarial after being filtered with the class-level essential frequency mask.

Table 4: Attack success rate for FMN, PGD and DeepFool when adversarially perturbed images are directly fed into the ResNet-20 classifier (A) and when those images are filtered using class-level essential frequency masks (B).

| FMN | | PGD | | DeepFool | |
|---|---|---|---|---|---|
| A | B | A | B | A | B |
| 99.96% | 7.89% | 98.77% | 7.90% | 100.00% | 7.85% |

## A.8 CORRELATION WITH DIFFERENT ADVERSARIAL PERTURBATION MAGNITUDES

The PGD attack algorithm allows to tune the parameter $\epsilon$, which controls the magnitude of the adversarial perturbation in terms of $\ell_\infty$ norm. Throughout the paper, we kept this value fixed at 0.01, to maintain consistency with the perturbation sizes found by the other attacks we employed. However, varying this parameter can provide useful insight into our results in terms of correlation between essential frequency masks and adversarial frequency masks. Interestingly, as shown in Table 5, the correlation does not directly depend on the adversarial perturbation being small, and it emerges with high confidence even with a stronger attack ($\epsilon = 0.03$), whose success rate is $100\%$.

Table 5: Attack success rates and correlation P-values for PGD with variable perturbation magnitude $\epsilon$ on ResNet-20.

| $\epsilon$ | Attack success rate | P-value |
|---|---|---|
| 0.005 | 82.19% | $3.2 \cdot 10^{-6}$ |
| 0.01 | 98.77% | $5.4 \cdot 10^{-8}$ |
| 0.03 | 100.00% | $2.1 \cdot 10^{-3}$ |

