# OpenReview forum: "Relating Implicit Bias and Adversarial Attacks through Intrinsic Dimension"
_ICLR.cc/2024/Conference — Submitted to ICLR 2024_

### Official Review · Reviewer_3jNL · 2023-10-24

**Soundness:** 3 good
**Presentation:** 3 good
**Contribution:** 2 fair
**Rating:** 3
**Confidence:** 3

**Summary:**

The focus in this paper is on minimum one-norm frequency masks of images with respect to a given neural network classifier.  They are such that if the fast Fourier transform of an image is multiplied pixel-wise by the mask and then the inverse of the fast Fourier transform is applied, the resulting image is assigned the same class by the network as the original image.  The authors propose a technique based on the notion of intrinsic dimension to measure possibly non-linear correlation, and apply it to assess correlation between the frequency masks of images and their adversarial perturbations.  The main result is that some correlation is shown in this way for CIFAR-10 and Imagenette datasets.

**Strengths:**

The figures in the paper help with understanding.

The authors indicate interesting directions for future work in the last section.

**Weaknesses:**

The reported decreases in the estimates of intrinsic dimension compared with the randomly shuffled data are relatively small.

One would expect some correlation between the frequency masks of the images and their adversarial perturbations simply because the adversarial noise is small.  I am not sure what beyond that I can conclude from the main result in the paper.

The proposed technique for assessing possibly non-linear correlation makes sense, however this paper shows that it is currently not practical for high-dimensional data, e.g. already for the Imagenette dataset there are computational and reliability issues.

The lack of space between paragraphs makes the paper harder to read, and is contrary to the formatting instructions in the LaTeX template for the conference.

**Questions:**

What classes do the adversarial attacks used in the paper target, this does not seem to be stated anywhere in the paper?

---

> ### Author Response · Authors · 2023-11-16
>
> We thank the Reviewer for their comments on our work, and respond to the weaknesses and questions that were outlined in the review.
>
> **On the small decreases in intrinsic dimension:** It is true that in some cases the decreases in the intrinsic dimension (ID) compared to the shuffled data sets are relatively small. This is the reason why we decided to adopt a statistical test to assess the significance of such variations.
>
> **Correlation is expected:** We agree that a strong connection between the implicit bias and the adversarial vulnerability of a neural network was to be expected. However, we point out that, to the best of our knowledge, this work is the first one to explicitly pinpoint such a connection. Besides, our result is not limited to prove the correlation: we give a very precise image-specific map of the spectral implicit bias. This information is potentially useful to design new strategies for adversarial defense and interpretable networks, topics which we are addressing as future goals. Moreover, motivated by this comment and by points raised by other Reviewers, we decided to test our approach on adversarial attacks with higher perturbation budget (namely, PGD with $\epsilon=0.03$). Our findings are not harmed by this choice, as shown in the revised pdf (Appendix, section A.8). Thus, the correlation is not solely justified by the fact that adversarial noise is small.
>
> **Issues of the correlation method:** While it is clear that some limitations emerge in the case of very high-dimensional data and/or very high-ID data (the case of Fourier masks on Imagenette is particularly challenging as it falls into both categories), we believe that the range of feasible applications of our ID-based correlation method is still very broad.
>
> **Paragraph spacing:** We thank the Reviewer for pointing out this issue, we fixed the paragraph spacing in the new version of our manuscript.
>
> **Question:** All attacks are used in their untargeted version, meaning that the only goal of the attack is to produce misclassification, regardless of the adversarial class. We thank the Reviewer for giving us the opportunity to clarify this point in the revised pdf file.

---

> > ### Comment · Reviewer_3jNL · 2023-11-16
> >
> > Thank you for these responses.

---

### Official Review · Reviewer_avqz · 2023-10-28

**Soundness:** 2 fair
**Presentation:** 2 fair
**Contribution:** 2 fair
**Rating:** 3
**Confidence:** 5

**Summary:**

The goal of this paper is to explore the connections between implicit bias and the adversarial vulnerability of deep neural networks.

**Strengths:**

The research topic in this paper is essential for the ML community.

**Weaknesses:**

The contributions are somewhat limited. In this paper, the authors do not provide sufficient evidence to support their arguments for the strong correlations between implicit bias and adversarial attacks. From Tab.2, the cosine similarity is not sharp. The authors should present many more results (such as the adversarial success rates against non-linear correlations) to clarify this. Moreover, the important thing is that the authors need to give a clear presentation about why we need to relate the implicit bias and adversarial attack together. Then, the proposed method is limited and incompleted in Sec.3.2, which is challenging to apply to practical applications.

How do we know the adversarial class? Since we usually adopt an untargeted adversarial attack, how do we compute Eq.(1) for adversarial examples? Also, the obtained frequency masks should be explained further. Compared to the model trained with original features, what is the performance if we only consider the mask features of images to train a model?

Finally, this paper does not propose new methods or strategies on top of their findings. What can we do next after finding the correlations?

**Questions:**

Please address all my concerns in the Weaknesses part.

---

> ### Author Response · Authors · 2023-11-16
>
> We thank the Reviewer for their feedback on our work. Please see our response below.
>
> **On the cosine similarity:** The fact that the cosine similarity between masks is not sharp is one of the crucial points of our work. We find that the correlation between sets of masks is highly non-linear and thus not detectable by cosine similarity. This is the reason why we designed and applied a novel non-linear correlation method based on intrinsic dimension estimation.
>
> **On additional results:** We thank the Reviewer for suggesting to inspect how correlations are related to adversarial success rates. We ran an experiment using PGD with variable perturbation budgets and found that correlation appears in any case. Results are available in the Appendix of the revised pdf (section A.8).
>
> **The proposed method is limited and incompleted:** We are aware that there are limitations to our ID-based correlation approach, mainly related to the computational issues of ID estimation in the case of very high-dimensional or high-ID data sets. We clearly state these limitations in the paper (sections 3.2 and 4.6). However, we do not fully understand what the Reviewer refers to with the term ‘incompleted’.
>
> **Why we need to relate the implicit bias and adversarial attack:** The problem of adversarial vulnerability of neural networks remains largely unresolved and, to a certain extent, not fully understood. We believe that our work represents a step towards understanding adversarial attacks and, potentially, defending against them.
>
> **On the adversarial class:** The Reviewer is right in saying that the attacks we employ are untargeted. We state this point explicitly in the revised version of our pdf file. The adversarial class is simply known by feeding the adversarial input to the classifier.
>
> **New methods or strategies on top of our findings:** We believe that this work can be a starting point towards new strategies for adversarial defence, as indicated in the future directions (section 5). Moreover, we sketch what we believe to be a promising idea in this direction in the last lines of the section that deals with class-specific masks (section 4.7).

---

### Official Review · Reviewer_hyn3 · 2023-10-30

**Soundness:** 1 poor
**Presentation:** 2 fair
**Contribution:** 2 fair
**Rating:** 3
**Confidence:** 4

**Summary:**

The primary objective of the paper is to find a correlation between the frequencies that a hypothesis is attending to correctly classify an image with the frequencies that adversarial attacks take advantage of. The proposal is that if such a correlation exists, then "the network spectral bias determines the nature of the adversarial attacks in the Fourier space".

To this end, the paper first provides a light review of the concept of "implicit bias" and its relationship with Fourier transform of images. The paper then follows to describe the issue of adversarial attacks and the concept of intrinsic dimension of a data set. Next, the methodology of the proposal is described in which a method of Karantzas et al. is used to obtain the aforementioned frequencies. The paper then describes a method for determining the existence of correlation between two sets of data based on the difference between the intrinsic dimension of the concatenation of the two sets with the intrinsic dimension of the concatenation of one of the sets with a random shuffling of the other set. The argument is supported by providing an example of a 2D spiral.

Finally, the paper reports the results of performing the proposed method on CIFAR and ImageNet data sets. The results suggest that the intrinsic dimension of concatenated natural and adversarial frequencies is on average increased by 4 or 5 almost surely when one the natural or adversarial samples is randomized.

**Strengths:**

### Originality:
The paper is original in its use of topological information to determine correlation between two sets of data.

### Quality:
The paper is well-written for the most parts.

### Clarity:
The language is simple and the paper refrains from using complicated mathematical concepts.

### Significance:
The paper proposes a method for solving the problem of efficient determination of correlation between subsets of $\mathbb{R}^n$ for $n >> 1$. This is a very well-known problem and is a subject of active research.

**Weaknesses:**

### Minor:
- The Z-test is not used in practice that much. Moreover, it is not appropriate if the statistics of the population is not known beforehand. I think that it should be replaced with a t-test at the very least. Moreover, the distribution of intrinsic dimension cannot be normal, as it is a positive number, so a Poisson/Gamma distribution is more appropriate. Even though the issue of the distribution does not appear to be detrimental to the results of the paper, the issues of Z-test should be clearly stated in the paper.

- The conventional $\epsilon$ for $\ell_\infty$ attacks of CIFAR is $\frac{8}{255} \approx 0.03$, whereas in the experiments it is $0.01$ (refer to https://robustbench.github.io/).

### Major:
- The proposal is poorly supported and the original concepts that are introduced in the paper does not have any rigorous or concrete definition.

- The literature review does not clear up what exactly "implicit bias" is, or how it is related to Fourier transform. The paper just assumes that the reader has the required background knowledge and only describes these relations as some "deep connection".

**Questions:**

- Please elaborate on the significance of the proposal from the perspective of robustness in machine learning. Specifically, I don't see the significance of providing "empirical evidence that the network bias in Fourier space and the target frequencies of adversarial attacks are closely tied". The literature on this issue is pretty big as evident from a simple search in Google Scholar.

- I don't find the verbal description of the proposed correlation test convincing at all. Even if the concept is sound, no formal statement or expression of the method is present. I cannot see how it is possible to verify the arguments made in the paper in its current form.

---

> ### Author Response · Authors · 2023-11-16
>
> We thank the Reviewer for their comments and for giving us the opportunity to improve the robustness of our work. We respond here to the weaknesses and questions raised in the review.
>
> **Minor weaknesses:**
>
> **1.** The Reviewer is right when assessing that the true distribution of the ID cannot be Gaussian, due to the impossibility of obtaining negative values. However, in all our tests, the distribution of the IDs computed after shuffling was nearly Gaussian, somehow justifying the choice of the Z-test. In any case, we coincide with the Reviewer's statement that, although the results of the paper are not affected by this issue, it should be at least commented on, so we added a sentence to the paper on the limitations of the Z-test (section 3.2).
>
> **2.** As we detail in the paper (section 4.3), we chose $\epsilon=0.01$ for PGD to maintain consistency with the perturbation sizes found by the other attacks. This choice does not harm the efficacy of the attack, which succeeds in almost 99% of the cases (please see Table 4, in the Appendix). However, motivated by this suggestion and by other points raised in the Reviews, we decided to test our method on other perturbation magnitudes, including $\epsilon=0.03$. The results confirm our findings and they are reported in the Appendix (Section A.8) in the revised version of our pdf.
>
> **Major weaknesses:**
>
> **1.** Please refer to the response to question 2.
>
> **2.** We could only provide a compact literature review due to space constraints. However, in section 2.1 we provide pointers to a number of papers that have analysed implicit bias in Fourier space (e.g. Rahaman et al., 2019 and Fridovich-Keil et al., 2022). The connection between implicit bias and adversarial attacks has not been as thoroughly investigated and it is, in fact, the focus of our work.
>
> **Questions:**
>
> **1.** We believe that understanding the phenomenon of adversarial vulnerability is a fundamental milestone towards building more robust and reliable models. As anticipated in the previous point, the literature on the relationship between implicit bias and adversarial attacks is, to the best of our knowledge, quite limited at the moment. We cite one paper by Faghri et al. (2021) as the main inspiration four our work. However, it is a purely theoretical paper and it deals with linear networks. Our work extends their results to more complex and useful models using an empirical approach.
>
> **2.** The strong connection between correlations among data features and the intrinsic dimension of a dataset is a widely acknowledged concept in the field of unsupervised learning. For instance, in the context of hyper-planar manifolds, these correlations manifest in the PCA spectrum, creating a discernible gap that establishes the correct projection dimension. In the case of non-linear manifolds, where correlations exhibit non-linear characteristics, this connection persists, although its mathematical definition becomes somewhat elusive and hinges on the specific method employed for determining the intrinsic dimension. Since our method is designed to be agnostic to the intrinsic dimension determination technique, we purposefully refrain from providing a detailed mathematical description, opting instead to showcase the intuitive aspects of the method and its application to both toy and realistic models. We hold the firm belief that the tests we present are sufficient to support the conclusions outlined in the paper. This confidence is rooted in our verification that artificial uncorrelated by-design datasets with the same macro-features as those investigated in the paper yield clearly different results than those that we found correlated.

---

> > ### Comment · Reviewer_hyn3 · 2023-11-18
> >
> > Thank you for the detailed response. However, I am not convinced that the paper is ready to be published.
> >
> > > We could only provide a compact literature review due to space constraints.
> >
> > The definition and the description of "implicit bias" is central to the argument of the paper. Even though its description would be written in the literature review, it is crucial for it to be present in the paper. IMHO, the paper should be reorganized so that the definitions could be placed in the main text.
> >
> > > although its mathematical definition becomes somewhat elusive and hinges on the specific method employed for determining the intrinsic dimension.
> >
> > While it is perfectly fine for a paper to be somewhat of an "idea proposal", I don't think that it is enough for a top conference or journal. The proposal of a theory-oriented paper would be valuable when it deals with the nuances of the mathematical definitions.

---

### Official Review · Reviewer_Zxej · 2023-11-03

**Soundness:** 2 fair
**Presentation:** 4 excellent
**Contribution:** 2 fair
**Rating:** 5
**Confidence:** 4

**Summary:**

This paper presents an empirical investigation of the correlation between the
essential input frequencies required for accuracy of neural network image classifiers, and the frequencies targeted by adversarial attacks. They follow a recent approach by Karantzas et al. in which a Fourier mask is computed for a given input image, and then compared (in the Fourier space) with a mask similarly computed for a successful attack of the image. The authors seek to test their assertion that the network spectral bias determines the nature of successful attacks. Following Karantzas et al., the modulatory masks are computed by applying a Fast Fourier Transform, and then masking with a matrix of traininable parameters between 0 and 1. The result is converted back into an image by taking the real part of the inverse FFT, and then fed to the pretrained classifier to obtain a prediction that is used to train the mask parameters.

The authors assess the relationships between original images and their successfully attacked counterparts through a test based on intrinsic dimensionality on a vector formed by flattening and concatenating the two masks. They reason that correlation between the two would be revealed by a strong drop in ID, as compared with that of vector in which the coordinates derived from one of the masks. Z-scores are used to test the hypothesis that the drop in ID is significant. The authors then present empirical evidence for their claims based on this test.

**Strengths:**

S1) The authors' observation that ID collapses when two sets of variables are highly correlated is an interesting. The contrast in ID between the cases of shuffled vs unshuffled variables is a compelling argument for the existence or lack of a correlation effect.

S2) For the image datasets and learning models considered in the empirical evaluation, the authors have made a good case for regarding the susceptibility of adversarial attack as being revealed by frequency components. The authors further exploit the idea of shuffling to demonstrate the existence of class-specific information in the masks.

S3) The paper presentation is of a high standard: well-organized, well-written, and clear.

**Weaknesses:**

W1) Among the technical contributions of the paper, the only novel idea is that of testing correlation using intrinsic dimensionality and variable shuffling. In other respects, the authors draw upon the Fourier mask framework of Karantzas et al. for analyzing the robustness of ANNs.

W2) The authors have not accounted for the potential for bias in their Z-score test. The variance of the ID (and therefore the significance of the Z-score) itself strongly depends on the dimensionality within which it is assessed. Setting a threshold for hypothesis testing that is uniform across all dimensions may not be appropriate here. Estimation of ID also has its own biases that may confound hypothesis testing of this type. Accounting for (and if necessary, adjusting for) dimensional bias would greatly improve both the importance and novelty of the results.

**Questions:**

Please address the point raised as W2.

---

> ### Author Response · Authors · 2023-11-16
>
> We thank the Reviewer for carefully reading our manuscript and for providing interesting points of discussion. We try here to address the weaknesses pointed out in the review.
>
> **W1:** The Reviewer correctly references Karantzas et al. (2022) as the main inspiration for our Fourier mask framework. However, we would like to note that, while the concept of essential frequency mask had already been explored in the aforementioned paper, the adversarial frequency masks are a novel contribution of our work.
>
> **W2:** Indeed, it is crucial to consider the variance associated with ID estimation. Employing a maximum-likelihood approach, one can obtain an expression for the variance and incorporate this insight as a correction term in the Z-score test. This would result in a slight modification of the p-value while maintaining unchanged conclusions. Additionally, addressing the bias in ID estimation could likely enhance the results and enable the extension of the test to datasets with a higher intrinsic dimension. However, tackling this remains an open problem in the unsupervised learning community, and its solution falls beyond the scope of this work. It is noteworthy, though, that the method proposed here remains independent of the ID estimator used (that is, if one had a perfect ID estimator, it would equally work on the method).  Moreover, by comparing measures of the ID estimated on the same embedding dimension, one could expect that many of the potential biases vanish – or, at least, are mitigated – by error compensation.

---

> > ### Comment · Reviewer_Zxej · 2023-11-23
> >
> > Thank you for your replies.

---

### Author Response · Authors · 2023-11-16

We thank the Reviewers for their comments and suggestions on our work. The main weaknesses and questions raised by each Reviewer are addressed in the individual replies.

We uploaded a revised version of our manuscript, containing the following modifications:

- Clarifications on the limits of the Z-test for non-linear correlation (section 3.2) and on the untargeted nature of the adversarial attacks we employ (section 4.3);
- Additional results on the correlation between essential frequency and adversarial frequency masks with variable perturbation magnitude (section A.8 in the Appendix);
- Fixed paragraph spacing.

---

### Meta-Review · Area_Chair_j92x · 2023-12-14

**Metareview:**

The paper investigates the relationship between the implicit bias of an NN architecture and adversarial attacks. Concerns about the novelty and the statistical procedures used (biases) were raised in the reviews. Some of these have been clarified in the discussion.

**Justification For Why Not Higher Score:**

NA

**Justification For Why Not Lower Score:**

NA

---

### Decision · Program_Chairs · 2024-01-16

Reject